# Donor Pericardial Interleukin and Apolipoprotein Levels May Predict the Outcome after Human Orthotopic Heart Transplantation

**DOI:** 10.3390/ijms24076780

**Published:** 2023-04-05

**Authors:** Éva Pállinger, Andrea Székely, Evelin Töreki, Erzsébet Zsófia Bencsáth, Balázs Szécsi, Eszter Losoncz, Máté Oleszka, Tivadar Hüttl, Annamária Kosztin, Edit I. Buzas, Tamás Radovits, Béla Merkely

**Affiliations:** 1Department of Genetics, Cell- and Immunobiology, Semmelweis University, 1085 Budapest, Hungary; 2Department of Anesthesiology and Intensive Therapy, Semmelweis University, 1085 Budapest, Hungary; 3Heart and Vascular Center, Semmelweis University, 1085 Budapest, Hungary; 4Faculty of Medicine, Semmelweis University, 1085 Budapest, Hungary; 5Doctoral School of Theoretical and Translational Medicine, Semmelweis University, 1085 Budapest, Hungary; 6HCEMM-SU Extracellular Vesicle Research Group, Semmelweis University, 1085 Budapest, Hungary; 7ELKH-SE Translational Extracellular Vesicle Research Group, Semmelweis University, 1085 Budapest, Hungary

**Keywords:** organ donor, heart transplantation, interleukin-6, apolipoprotein, brain death

## Abstract

The proinflammatory cascade that is activated at the time of brain death plays a crucial role in organ procurement. Our aim of this study was to explore the relationship between the clinical outcome of orthotopic heart transplantation, as well as cytokine and apolipoprotein profiles of the pericardial fluid obtained at donation. Interleukin, adipokine and lipoprotein levels in the pericardial fluid, as well as clinical data of twenty donors after brain death, were investigated. Outcome variables included primary graft dysfunction, the need for posttransplantation mechanical cardiac support and International Society for Heart and Lung Transplantation grade ≥ 2R rejection. Hormone management and donor risk scores were also investigated. Lower levels of IL-6 were observed in primary graft dysfunction (median: 36.72 [IQR: 19.47–62.90] versus 183.67 [41.21–452.56]; *p* = 0.029) and in the need for mechanical cardiac support (44.12 [20.12–85.70] versus 247.13 [38.51–510.38]; *p* = 0.043). Rejection was associated with lower ApoAII (*p* = 0.021), ApoB100 (*p* = 0.032) and ApoM levels (*p* = 0.025). Lower adipsin levels were detected in those patients receiving desmopressin (*p* = 0.037); moreover, lower leptin levels were found in those patients receiving glucocorticoid therapy (*p* = 0.045), and higher T3 levels were found in those patients treated with L-thyroxine (*p* = 0.047) compared to those patients not receiving these hormone replacement therapies. IL-5 levels were significantly associated with UNOS-D score (*p* = 0.004), Heart Donor Score (HDS) and Adapted HDS (*p* < 0.001). The monitoring of immunological and metabolic changes in donors after brain death may help in the prediction of potential complications after heart transplantation, thus potentially optimizing donor heart allocation.

## 1. Introduction

Orthotopic heart transplantation (HTX) has been used as an optimal treatment for end-stage heart failure. Strict selection criteria and the application of score systems for the determination of urgency status on the waiting list have further aided in decreasing adverse outcomes [1]. Despite improvements in medical and device optimization in recent decades, heart transplantations still have a high complication rate, including primary graft dysfunction (PGD), early graft loss and rejection [2].

Furthermore, optimal procurement after brain death also plays an important role in the outcome of HTX. Due to the organ shortage, expanded donor selection criteria have been used [3]. In these cases, the donor heart does not fulfill the standard donor selection criteria. Organ improvement includes hormone resuscitation and machine perfusion.

Systemic inflammatory reaction syndrome (SIRS) is triggered after brain death. This process initiates or increases the immune reaction [4]. The release of proinflammatory cytokines is followed by tissue infiltration of inflammatory cells [5]. After donor explantation, ischemia‐reperfusion injury further progresses with the immunological reaction and leads to different degrees of organ dysfunction (or even to death). Donor-related risk can be quantified by using different scoring systems, such as the United Network for Organ Sharing (UNOS) or the Eurotransplant developed scores [6,7]. In addition to the cold ischemia time, different donor-related factors, such as diabetes or age, are included among the variables.

We hypothesized that the extent of immunological injury can be quantified and that we can obtain an in-depth assessment of heart function if pericardial fluid samples are analyzed. The aim of this study was to explore the relationship between the cytokine and apolipoprotein (Apo) profiles in the pericardial fluid of the donor heart and the clinical outcomes of orthotopic HTX, including PGD, organ rejection and mortality. Pericardial hormone concentrations were also determined in relation to hormone management and donor risk scores.

## 2. Results

### 2.1. Donor Characteristics

The median age of the donors was 30 years, and 5 donors (25%) were women (n = 5). There was gender mismatch in 15% (n = 3) of the transplantations, and the medians of the donors’ and recipients’ body mass index (BMI) values were 25.40 and 25.65 kg/m^2^, respectively. In most of the cases, the cause of brain death was trauma (n = 16, 80%) with or without subdural hematoma. Further descriptive characteristics of the donors are summarized in Table 1.

### 2.2. Relationship between Donor Risk Scores and Immune Parameters

The median scores of the patients were 0 (IQR: 0–0.75), 16 (IQR: 15–21), and 3.77 (IQR: 3.69–4.38) for the UNOS-D, HDS, and adapted HDS scores, respectively. Each donor risk score correlated with the IL-17A and IL-5 levels. Apo I and Apo A II levels negatively correlated with adapted HDS scores. The UNOS-D score correlated with adiponectin levels. The data are shown in Table 2 and Appendix A.

### 2.3. Relationship between Donor Interleukin Levels and Postoperative Complications

After transplantation, six patients (30%) needed postoperative MCS, and five patients had PGD. Five patients had ISHLT Grade 2R rejection.

In the pericardial fluid of the donors, IL-6 levels were significantly lower in patients with PGD compared to those without PGD (*p* = 0.029). IL-6 levels were also significantly lower in the MCS group (*p* = 0.043). The detailed comparisons are shown in Table 3, Figure 1, and Appendix A. In the pericardial donor fluids, thiol levels were significantly higher in patients who had postoperative vasoplegia compared to those who had no vasoplegia (median: 6.54 [2.10–10.35] vs. 0.73 [0.00–3.17], *p*= 0.045 in the vasoplegic and the nonvasoplegic patients, respectively) (Appendix A). In patients who had rejection in the first month after transplantation (n = 5, 20%), apolipoprotein levels were higher compared to the nonrejection group. The data are shown in Table 4 and Appendix A.

Donor age was higher in the PGD and rejection analyses among those who suffered these adverse events compared to those who did not (*p* = 0.044 and *p* = 0.018 in the PGD and rejection analyses, respectively). Recipient UNOS scores were not different in the investigated outcomes. We have found that ApoM (r: −0.674, *p* = 0.001), ApoAII (r: −0.613; *p* = 0.004) correlated with the age of the donors. The data are shown in Appendix A.

### 2.4. Effects of Hormone Replacement Therapy on Interleukin Levels

Out of 20 donors, 6 donors (30%) did not receive any replacement, 7 donors (35%) received 1 hormone replacement, 2 patients (10%) received 2 types of hormone therapy and 5 donors (25%) had desmopressin-steroid and L-thyroxin replacement 48 h before explantation. The adipsin level was significantly lower in donors who received desmopressin therapy compared to those who did not have desmopressin replacement therapy (median: 492.51 [425.77–538.68] ng/mL vs. 588.77 [540.54–627.24] ng/mL, *p* = 0.037 in desmopressin-substituted and in not substituted patients, respectively). Moreover, the administration of hydrocortisone or methylprednisolone was associated with lower leptin levels in comparison to those who were not treated with glucocorticoid (median: 4.60 [4.18–5.24] ng/mL vs. 5.19 [4.79–5.60] ng/mL, *p* = 0.045 in patients who received and who did not receive glucocorticoid therapy, respectively). Leptin levels had no significant correlation with body mass index (r = 0.120, *p* = 0.613). In L-thyroxine-substituted patients, higher pericardial fluid T3 levels were measured compared to those who were not treated with L-thyroxine (median: 2.46 [1.95–2.88] pg/mL vs. 1.99 [0.43–2.37] pg/mL, *p* = 0.047, in patients who received and who did not receive L-thyroxine replacement therapy, respectively). No significant correlation was found between the number of substituted hormones and interleukin levels (r = 0.120, *p* = 0.613). (Figure 2, Appendix A).

### 2.5. Laboratory and Echocardiography Parameters

Significant correlations were found between several immunological and laboratory or echocardiography parameters. The detailed results are shown in Appendix A.

## 3. Discussion

We found that the donor pericardial IL-6 level was lower in the transplanted patients who had PGD or MCS support than in those patients who had not received support. Patients who had rejections had significantly higher apoprotein levels. In the case of hormone replacement, desmopressin therapy caused lower adipsin levels, and steroid treatment was associated with lower leptin levels. IL-17A and IL-5 levels correlated with the UNOS, HDS and aHDS donor scores.

Brain death has been shown to cause a significant acceleration of immunological reactions, which is characterized by the expression of cytokines and adhesion molecules, followed by the infiltration of leukocytes [16]. The resulting inflammation can lead to higher immunogenicity and acute rejection. In experimental conditions testing brain death rats, acute rejection after brain death donation was more intense and rapid compared to living donor controls [16]. Our findings also support the concept of the accelerated immune response in the brain-dead donor heart measured in the pericardial fluid. We obtained our samples from the pericardial fluid, supposing that the most relevant immunological responses can be detected in close proximity to the organ. The upregulation of IL-6 has been consistently observed in brain death patients [17,18]. In one study, the IL-6 elevation in brain death patients was similar to the elevation in septic patients [18]. IL-6 release can be explained by brain damage, but the additional role of sepsis during ICU treatment in certain patients cannot be excluded [19].

The main goal of HTX is to decrease mortality and to secure an improved quality of life after the procedure. Primary graft dysfunction still has a high occurrence rate and is the leading cause of early transplantation-related mortality [20]. The definition of PGD is low cardiac output and high filling pressures without signs of hyperacute rejection and recipient-related pulmonary hypertension. Several factors have been observed in association with PGD, such as ischemic time, retransplantation and undersized donors [21]. Techniques in organ preservation have been improved, but the mortality and morbidity after HTX still have not reflected this improvement. The occurrence of PGD varies between 20 and 40%, and it has not been shown to decline with time [22]. The vast majority of patients are operated on from the high urgency list requiring extracorporeal membrane oxygenation or ventricular assist device treatment. The donor shortage causes the trend that donors designated as “marginal donors” will also be accepted. Therefore, additional measurements of quantitative indicators, such as cytokines or interleukins, may further help in proper matching and donor heart selection [23]. We observed significantly lower IL-6 levels in cases of PGD. IL-6 is a pleiotropic, multifunctional cytokine that coordinates both innate and adaptive immune responses. Although IL-6 is known as a proinflammatory cytokine, it has both pro- and anti-inflammatory characteristics and has a role in numerous protective and regenerative procedures [24]. IL-6 family signaling seems to be protective in heart tissue in the first acute response, whereas a chronically elevated cytokine level leads to cardiac hypertrophy and depressed cardiac function [25]. IL-6 family signaling seems to have a crucial protective role in ischemia‐reperfusion injury [26]. The Janus-faced nature of the IL-6 cytokine and signaling may explain our results.

In recent decades, the UNOS and the Heart Donor Score have been developed for a better quantitative description of donor hearts [6,7,10]. We also calculated these scores and compared them to the immunological parameters. Evidently, important factors, such as total ischemic time, as well as donor-recipient mismatch in size or in gender, cannot be investigated from the samples obtained before explantation. Nevertheless, IL17A and IL5 interleukin levels, as well as lower apoprotein levels, were correlated with increasing risk severity scores. Higher adiponectin levels were associated with an increased risk of cardiovascular morbidity and mortality [27]. These markers can be markers of a complicated posttransplantation course, even in individuals with apparently healthy metabolic and cardiovascular parameters.

Hormone replacement and/or hormone resuscitation can be an important treatment tool for donor optimization [28]. After brain death, the absence of antidiuretic hormone causes hemodynamic instability via severe polyuria, hypovolemia and dehydration. The early replacement of desmopressin or continuous vasopressin infusion is indicated at each donor after brain death. Moreover, the administration of glucocorticoids and/or hydrocortisone is also recommended to achieve hemodynamic improvement [29]. The role of thyroid hormone replacement has been conversely discussed. Thyroxine replacement was associated with early graft loss and 30-day mortality by analyzing data from 23,000 HTX recipients [30]. The withdrawal phenomenon has been hypothesized in this process as after implantation the heart will be placed in a relative thyroid hormone deficiency of the recipient caused by sick euthyroid syndrome, amiodarone treatment and glucocorticoid treatment [31]. Reliable data from prospective studies are still missing regarding the benefit of thyroid hormone replacement. Further studies are also needed to clarify the role of substitution or the impact of actual hormone levels during procurement [32]. Thyroxine replacement was associated with higher levels of T3 in the pericardial fluid (but without any immunological modulation) in our donor population. In our study, desmopressin replacement decreased adipsin levels, which can be explained by an indirect effect through the adipocitokine network influencing glucose and lipid homeostasis [33]. Glucocorticoid treatment decreased the leptin level in our study population, which contradicts the previous findings that the administration of different steroids elevated leptin levels [34].

In addition to PGD, rejection is a feared complication after transplantation. Due to immunotherapy and established monitoring guidelines, the number of acute heart rejections has slightly decreased in recent decades [35]. Rejections were diagnosed by using endomyocardial biopsies in the first month after transplantation. Patients who had acute cellular reaction (ISHLT ACR II) had higher apolipoprotein levels in the donor pericardial fluid. Apolipoprotein A1 levels were a marker of biopsy-confirmed renal allograft rejection [36]. This study also demonstrated that lipoprotein metabolism may play a role in rejection. Additionally, we have found a strong negative correlation among the age of the donors and certain apolipoprotein levels, which might serve as additional markers in the rejection. In our study, higher donor age and high apolipoprotein levels in the pericardial fluid were associated with rejection. The explanation of this finding is further complicated by the fact that rejection occurs after transplantation, and immunosuppressive therapy can modulate the reaction. The monitoring of T-cell activity before and after transplantation may help in the early recognition of sample rejection [37].

### Limitations of the Study

The main limitation of this study was that the pericardial fluid samples of the recipients were not available. Therefore, the bilateral interactions could not be investigated in this study. Serum samples were not obtained in parallel with the pericardial fluids. Moreover, the small population size and the heterogeneity of the donors resulted in lower statistical power. However, a large number of cytokines and lipoproteins were measured, and the samples were obtained from the pericardium, which is in close proximity to the donor organ. The donor population represented the brain death patients procured by the national guidelines. Authors should discuss the results and how they can be interpreted from the perspective of previous studies and of the working hypotheses. The findings and their implications should be discussed in the broadest context possible. Future research directions may also be highlighted.

## 4. Materials and Methods

### 4.1. Study Design, Setting and Participants

Well-characterized pseudonymized human pericardial fluid and cell-depleted pericardial fluid samples of 20 randomly-chosen donors from the time period between February 2013 and December 2017 were obtained from the Transplantation Biobank of the Heart and Vascular Center at Semmelweis University, Budapest, Hungary [8]. Between January 2013 and April 2017, 206 orthotopic transplantations were performed and 121 donor pericardial fluid specimens were available (58.7% of the transplantations). The collection of the donor pericardial fluid is complicated by the fact that blood contamination of the pericardial fluid makes the analysis impossible. Among these samples, 20 samples were randomly selected and analyzed in the Department of Genetics, Cell- and Immunobiology. The procedure of sample procurement was reviewed and approved by the institutional and national ethics committee (ethical permission numbers: ETT TUKEB 7891/2012/EKU [119/PI/12.] and ETT TUKEB IV/10161-1/2020). Clinical patient data were obtained from the database of the Transplantation Biobank. Our study was conducted in accordance with the Eurotransplant standards for organ sharing and with the Hungarian National Blood Transfusion Service. The last check on the follow-up data was made on 30 October 2022 [9].

### 4.2. Local Protocols, and Donor Management

Donor and recipient variables were retrieved from the National and Eurotransplant-based donor data report form and from electronic medical records from our institutional databank. The following data on donors were collected and analyzed: age, sex, height, weight, body mass index, cause of brain death, overall length of stay (LOS), donor management time at the intensive care unit (ICU), diabetes mellitus, hypertension, smoking, drug abuse, active malignant tumor, serum sodium, serum potassium, serum chloride, serum glucose concentration, blood urea nitrogen (BUN), blood group, urine output (mL/kg/h), administration of inotropic and vasoactive medications (norepinephrine, epinephrine and dopamine), ejection fraction of the heart, interventricular septum thickness in end-diastole and left and right atrial and ventricular diameters. Hormone replacement therapy (hydrocortisone, methylprednisolone, thyroxine, desmopressin and vasopressin) administered in the last 48 h before procurement was also added to the variables. Patients who received more than one dose of hydrocortisone or methylprednisolone before procurement were considered as treated patients. Description of the study is shown in Figure 3.

### 4.3. Definitions and Measurements (Variables, Data Sources and Grouping)

Due to the relatively small sample size of our study, the UNOS score was calculated for donors, recipients and overall individuals. The donor-specific UNOS (UNOS D) score includes donor age (1 point if age is between 50 and 55 years; 2 points if age is above 55 years), total ischemic time (above 4 h, 2 points), sex mismatch (1 point), and donor diabetes mellitus (1 point). According to these criteria, the UNOS-D score was calculated, and three UNOS-D risk groups were formed: low (score: 0), intermediate (score: 1 or 2), and high (score: ≥3). The recipient-specific UNOS score considered the following parameters: age (above 65 years, 1 point), body mass index (30–35 kg/m2, 1 point; >35, 2 points), mean pulmonary artery pressure (above 30 mmHg, 1 point), total bilirubin (between 1.5 and 1.9 mg/dL, 1 point; >1.9 mg/dL, 2 points), creatinine (1.5–2.0 mg/dL, 1 point; >2 mg/dL, 2 points), previous transplant, previous cancer, and pretransplant mechanical ventilation (each 2 points) or mechanical circulatory support (non-continuous-flow 2 points) [7]. The total score is the sum of donor- and recipient-specific scores, and this score was used in the multivariable logistic regression analyses for adjustment.

#### The Heart Donor Score and the Adapted Heart Donor Score

The original Heart Donor Score (HDS) includes the following 12 donor characteristics: age, cause of death (cranial trauma, benign brain tumor, malignant brain tumor, circulatory disorders, cerebrovascular accident (CVA), drug overdose, intoxication, carbon monoxide intoxication, meningitis, respiratory disorders, subarachnoid bleeding and sepsis), compromised history (histories of drug abuse, malignancy, sepsis, meningitis, positivity for hepatitis B surface antigen, hepatitis B core antibodies or hepatitis C virus antibodies), hypertension, cardiac arrest, left ventricular ejection fraction (LVEF), valve function, left ventricular hypertrophy (LVH), coronary angiography, serum sodium, norepinephrine support and dopamine/dobutamine support [6]. The adapted Heart Donor Score (aHDS) used the original HDS (except for the variables of valve function, left ventricular hypertrophy and history of drug abuse). Beta estimates (log odds ratios) of this multivariable regression model were then used as points for the calculation of the aHDS, and they were summarized and expressed as aHDS [10].

### 4.4. Sample Collection and Preparation

After sternal splitting and opening of the pericardium, ACD (Tri-sodium Citrate with Citric Acid and Dextrose) vacutainer tubes (BD Vacutainer^®^ System, BD Biosciences, San Jose, CA, USA) were used for the collection of pericardial fluid samples. Cells were removed via centrifugation (300 g, 10 min, 40 °C). The cell-free samples were recentrifuged (2000 g, 10 min, 40 °C), and these supernatants were divided into three 500 µL aliquots and stored in liquid nitrogen until use.

### 4.5. Flow Cytometric Multiplexed Bead-Based Immunoassays

LEGENDplex™ bead-based immunoassays (BioLegend, San Diego, CA, USA) were used for the quantification of pericardial fluid cytokines, adipokines and lipoproteins according to the manufacturer’s instructions.

### 4.6. Enzyme-Linked Immunosorbent Assays (ELISA)

Pericardial fluid oxidized low-density lipoprotein (oxLDL), triiodothyronine (T3) and thyroxine (T4) concentrations were determined via ELISA (human OxLDL ELISA^®^ Kit of Cloude Clone; T3 and T4 Human ELISA Kits Abcam, Cambridge, UK) according to the manufacturer’s instructions. Oxidative damage to pericardial fluid proteins was assessed by determining the free SH content expressed in cysteine equivalents via the DTNB (5,5′-dithio-bis-(2-nitrobenzoic acid = Ellman reagent) method (Thermo Scientific Pierce Ellman’s Reagent).

### 4.7. Outcomes

Our primary outcome was the primary graft dysfunction in recipients. PGD was defined according to the consensus criteria of the International Society of Heart and Lung Transplantation (ISHLT) [11]. Severe PGD is the most common cause of short-term mortality after HTX [12,13]. The decision of mechanical circulatory support (MCS) implantation was made by a team of experts (including a cardiologist, a cardiac surgeon and a cardiac anesthesiologist) according to international guidelines [14]. Acute rejection was defined as an event that necessitated increased immunosuppression with an ISHLT grade ≥ 2R endomyocardial biopsy result or with noncellular reactions with hemodynamic compromise. Patients underwent routine surveillance for allograft function via endomyocardial biopsy and echocardiography at 1, 2, 3 and 4 weeks, as well as at 3, 6, 9 and 12 months after transplantation. Posttransplantation vasoplegia was defined according to the binary definition of vasoplegia: cardiac index of 2.5 L/min/m2 or greater and the need for noradrenaline (≥5 µg/min), adrenaline (≥4 µg/min) or vasopressin (≥1 unit/h) to maintain a mean arterial blood pressure of 65 mm Hg for 6 consecutive hours during the first 48 h after surgery [15].

### 4.8. Statistical Analysis

Data are expressed as the median and interquartile range (first–third [IQR]). Differences between the groups were assessed by the nonparametric Mann‐Whitney U test for continuous variables. The association between outcomes and inflammatory variables was tested via Spearman’s correlation. Comparisons between interleukins (IL), apolipoproteins and donor scores were conducted by performing a one-way analysis of variance on the ranks test. Data were analyzed by using IBM-SPSS 22.0 software (International Business Machines Corporation, Armonk, New York, NY, USA). All of the statistical tests were two-sided, and a *p* value < 0.05 was considered to be statistically significant.

## 5. Conclusions

In conclusion, IL-5 levels correlated with the UNOS, HDS and adapted HDS donor scores. PGD and the need for MCS were associated with lower IL-6 in the donors before explantation. Our results indicate that immunological and metabolomics markers should be measured to obtain more information about the apparently well-conditioned donor heart. The measurement is inexpensive and can be easily performed in the laboratory before transplantation. More in-depth knowledge of the immunological and metabolic changes after brain death may help in treatment optimization and donor procurement.

## Figures and Tables

**Figure 1 ijms-24-06780-f001:**
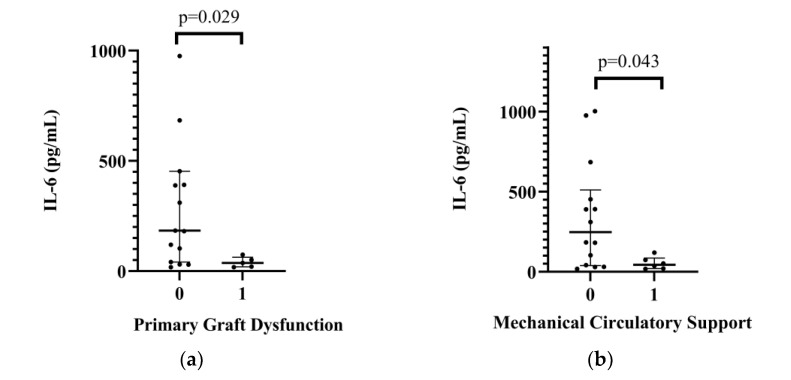
(**a**) Relationship between donor’s pericardial interleukin-6 levels (IL-6) and primary graft dysfunction after heart transplant. (**b**) Relationship between donor’s pericardial interleukin-6 levels (IL-6) and postoperative mechanical circulatory support after heart transplant.

**Figure 2 ijms-24-06780-f002:**
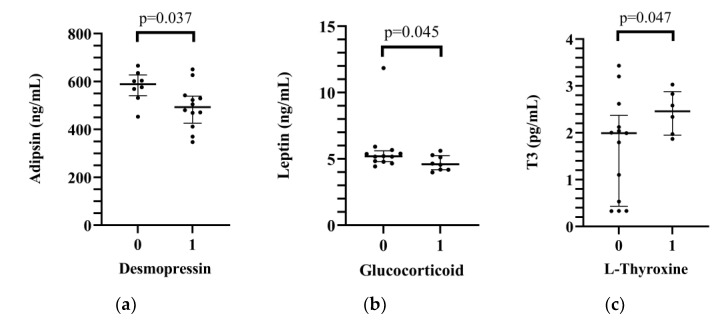
Relationship between donors’ desmopressin (**a**), glucocorticoid (**b**) and L-thyroxine (**c**) replacement therapy and the pericardial immune profile.

**Figure 3 ijms-24-06780-f003:**
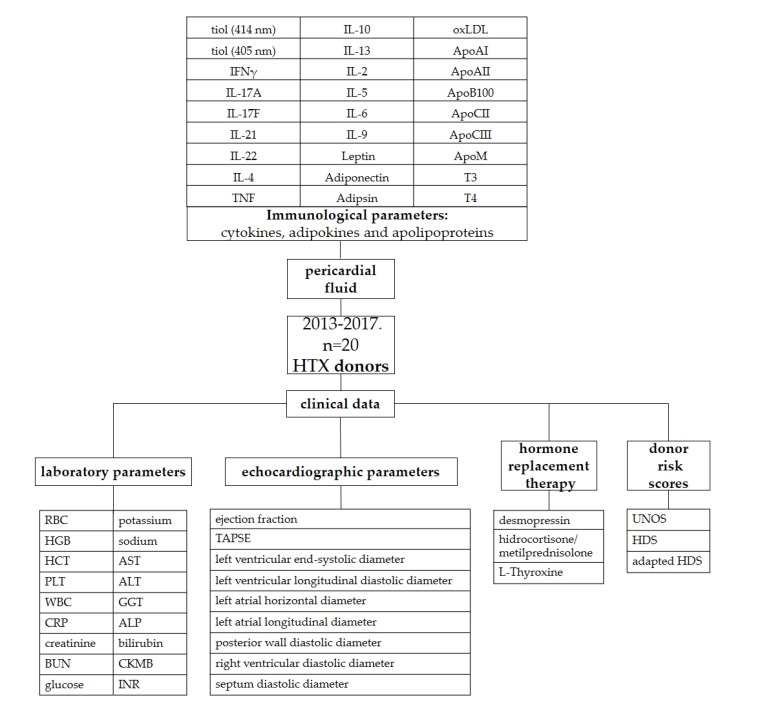
Description of the study. ALP: alkaline phosphatase; ALT: alanine transaminase; Apo: apolipoprotein; AST: aspartate aminotransferase; BUN: blood urea nitrogen; CKMB: creatin kinase-MB isoform; CRP: C-reactive protein; GGT: gamma-glutamyl transferase; HCT: hematocrit; HDS: Heart Donor Score; HGB: hemoglobin; HTX: heart transplantation; IFN γ: interferon-γ; IL: interleukin; INR: international normalised ratio; oxLDL: oxidized low-density lipoprotein; PLT: platelet count; RBC: red blood cell; T3: triiodothyronine; T4: thyroxine; TAPSE: tricuspid annular plane systolic excursion; TNF: tumor necrosis factor; UNOS: United Network for Organ Sharing; WBC: white blood cells.

**Table 1 ijms-24-06780-t001:** Descriptive characteristics of the donors and heart transplantations.

Factor	n Median	% (IQR 25–75)
BMI (kg/m^2^)	25.40	(22.05–27.78)
Age (year)	30.00	(23.25–37.00)
Cause of death		
Head trauma	16	80
Subarachnoid hemorrhage	2	10
Cardiac arrest	1	5
Intoxication	1	5
UNOS donor score	0	(0–0.75)
UNOS donor risk group		
low (0)	15	75
intermediate (1.2)	5	25
high (≥3)	0	0
HDS	16.00	(15.00–21.00)
Adapted HDS	3.77	(3.69–4.38)
Donor treatment		
Desmopressin	12	60
Hydrocortisone/Methylprednisolone	8	40
L-Thyroxine	6	30
Noradrenalin dose (µg/kg/min)	0.17	(0.09–0.26)
Dopamin/dobutamine dose (µg/kg/min)	0.00	(0.00–0.00)
Laboratory values		
Sodium (mmol/L)	144.00	(140.00–153.00)
Potassium (mmol/L)	4.00	(3.86–4.20)
Creatinine (µmol/L)	76.50	(63.25–96.25)
BUN (mmol/L)	4.40	(3.00–7.20)
Glucose (mmol/L)	8.30	(6.20–9.50)
CRP (mg/L)	133.70	(48.58–170.43)
CKMB (UI/L)	36.00	(11.00–97.00)
AST (UI/L)	56.00	(37.00–194.00)
ALT (UI/L)	44.00	(23.00–90.00)
GGT (UI/L)	28.00	(15.50–53.00)
ALP (UI/L)	140.00	(96.50–176.00)
Echocardiography parameters		
Ejection Fraction	62.00	(59.00–65.00)
TAPSE	22.00	(21.00–26.50)
Posterior Wall Diastolic Diameter	11.00	(10.00–11.50)
LVLDD	44.50	(42.00–48.00)
LVLSD	30.00	(26.75–33.50)
Left Atrial Longitudinal Diameter	33.50	(29.75–37.25)
Left Atrial Horizontal Diameter	37.00	(32.75–40.75)
Abnormal Valve Function	3.00	15.00
Transplant characteristics		
Total ischemic time (min)	185.00	(128.50–213.25)
Sex mismatch	3	15
Donor cardiac arrest	5	25

ALP: alkaline phosphatase; ALT: alanine transaminase; AST: aspartate aminotransferase; BMI: body mass index; BUN: blood urea nitrogen; CKMB: creatin kinase-MB isoform; CRP: C-reactive protein; HDS: Heart Donor Score; GGT: gamma-glutamyl transpeptidase; LVLDD: left ventricular longitudinal diastolic diameter; LVLSD: left ventricular end-systolic diameter; TAPSE: tricuspid annular plane systolic excursion; UNOS: United Network for Organ Sharing.

**Table 2 ijms-24-06780-t002:** Relationship between UNOS Donor Score as continuous variable, Heart Donor Score, adapted Heart Donor Score and immune parameters.

	Median	IQR (25–75)	UNOS-D *p* Value	HDS *p* Value	aHDS *p* Value
IL-17A (pg/mL)	8.56	(8.49–8.72)	0.004	<0.001	0.009
IL-5 (pg/mL)	1.04	(0.93–1.31)	0.004	<0.001	<0.001
Adiponectin (ng/mL)	16.07	(12.18–23.55)	0.006	0.030	
Leptin (ng/mL)	5.14	(4.58–5.39)	0.014		
ApoAI (ug/mL)	51.85	(41.52–94.00)			0.005
ApoAII (ug/mL)	194.83	(108.36–230.30)			0.030
ApoCII (ug/mL)	9.28	(8.40–10.06)			0.022
T4 (ug/dL)	3.26	(2.84–3.78)		0.002	

Apo: apolipoprotein; aHDS: adapted Heart Donor Score; HDS: Heart Donor Score; IL: interleukin; IQR: interquartile range; T4: thyroxine; UNOS-D: United Network for Organ Sharing Donor score.

**Table 3 ijms-24-06780-t003:** Relationship between donor’s pericardial interleukin-6 and postoperative complications.

	**No PGF**		**PGF**		**No MCS**		**MCS**	
IL-6 (pg/mL)	Median	IQR (25–75)	Median	IQR (25–75)	Median	IQR (25–75)	Median	IQR (25–75)
	183.67	(41.21–452.56)	36.72	(19.47–62.90)	247.13	(38.51–510.38)	44.12	(20.12–85.70)
*p* value	0.029				0.043			

IL: interleukin; IQR: interquartile range; MCS: mechanical circulatory support; PGF: primary graft dysfunction.

**Table 4 ijms-24-06780-t004:** Relationship between donor’s pericardial apolipoprotein levels and ISHLT Grade ≥2R rejection.

	No Rejection		Rejection		
	Median	IQR (25–75)	Median	IQR (25–75)	*p* Value
ApoAII (ug/mL)	177.35	(101.55–212.41)	339.08	(180.50–371.39)	0.021
ApoB100 (ug/mL)	65.01	(47.00–183.16)	358.92	(95.16–1570.34)	0.032
ApoM (ug/mL)	10.44	(0.00–17.91)	34.07	(18.57–180.63)	0.025

Apo: apolipoprotein; IQR: interquartile range.

## Data Availability

The data presented in this study are available on request from the corresponding author. The data are not publicly available due to privacy reasons.

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
