# Peer review of "Donor Pericardial Interleukin and Apolipoprotein Levels May Predict the Outcome after Human Orthotopic Heart Transplantation"

_ijms, 2023, doi:10.3390/ijms24076780_

Round 1

Reviewer 1 Report

In this interesting paper, the authors explored the relationship between the clinical outcomes of orthotopic heart transplantation, and cytokine and apolipoprotein profiles of the pericardial fluid obtained at donation. They observed lower levels of IL-6 in primary graft dysfunction and in patients needing for mechanical cardiac support. Moreover, rejection was associated with lower ApoAII, ApoB100, and ApoM levels, and different levels of adipsin, leptin and T3 were detected in patients receiving different hormone replacement therapies. Finally, IL-5 levels were significantly associated with UNOS-D score, Heart Donor Score (HDS) and Adapted HDS.

I think that, in the donor shortage era, every effort to improve donor management and to understand the mechanisms of PGD and rejection is important. Nevertheless, I think this paper could be improved. Here my suggestions:

1.     The paragraph order must be changed. The methods section needs to be discussed before the results, otherwise the paper is impossible to undestand

2.     What is “UNOS-D”? In paragraph 2.2 you state that the median UNOS-D score was 0 (IQR: 0-0.75), but you just reported in table 2 UNOS donor risk groups. 

3.     Page 4, line 108-109. “In the pericardial fluid of the donors, IL-6 levels were significantly lower compared to those without PGD (p=0.029).” I think the English of this sentence is wrong. You should specify that IL-6 levels were significantly lower in patients with PGD compared to those without PGD

4.     Page 4, line 110-113. “In vasoplegic patients, thiol levels were significantly higher compared to those who had no vasoplegia (median: 6.54 [2.10-10.35] vs 0.73 [0.00-3.17], p= 0.045 in the vasoplegic and the nonvasoplegic patients, respectively)”. You are probably talking about the donors but it not clear. Please specify.

5.     Moreover, how did you define vasoplegia in these patients? And did you analyze the effect of vasoplegia or thiol levels on post-transplantation outcomes? Otherwise I think that this information is useless in the paper and you should consider deleting it

6.     In the paragraph 2.4 “Effects of hormone replacement therapy on interleukin levels” you studied the effects of hormone replacement therapy on the levels of leptin, adipsin and T3. Why didn’t you you study the effect of these therapies on on IL-6 levels and apolipoprotein levels that have a clinical correlation with PGD or rejection? Have the levels of leptin, adipsin and T3 any correlation to clinical outcomes after heart transplantation? 

Author Response

The authors would like to thank Reviewer 1 for all the comments.
Please see attached the answers and the revision.

Reviewer 2 Report

In this study, the authors aimed to explore the relationship between the cytokine and apolipoprotein (Apo) profiles in the pericardial fluid of the donor heart and the clinical outcomes of Orthotopic heart transplantation (HTX), including primary graft dysfunction (PGD), organ rejection and mortality. Pericardial hormone concentrations were determined in relation to hormone management and donor risk scores.

Lower levels of IL-6 were observed in primary graft dysfunction (p=0.029) and in the need for mechanical cardiac support (p=0.043). Rejection was associated with lower ApoAII (p=0.021), ApoB100 (p=0.032) and ApoM levels (p=0.025). Lower adipsin levels were detected in those patients receiving desmopressin (p=0.037); moreover, lower leptin levels were found in those patients receiving glucocorticoid therapy (p=0.045), and higher T3 levels were found in those patients treated with L-Thyroxine (p=0.047), compared to those patients not receiving these hormone replacement therapies. IL-5 levels were significantly associated with UNOS-D score (p=0.004), Heart Donor Score (HDS) and Adapted HDS (p<0.001).

Study design, setting and participants: the authors should expand this section and provide additional information on the selection criteria of samples, clinical characteristics of patients to properly evaluate the presence of counfunding factors potentially affecting study results.

Author Response

The authors would like to thank Reviewer 2 for all the comments.
Please see attached the answers and the revision.

Round 2

Reviewer 1 Report

Dear authors,

Thank you v very much for answering all my questions. I think now your article is suitable for publication.

Sincerely

Reviewer 2 Report

The revised manuscript can be now accepted.